# Muscle Sonography in Inclusion Body Myositis: A Systematic Review and Meta-Analysis of 944 Measurements

**DOI:** 10.3390/cells11040600

**Published:** 2022-02-09

**Authors:** Ramy Abdelnaby, Khaled Ashraf Mohamed, Anas Elgenidy, Yousef Tarek Sonbol, Mahmoud Mostafa Bedewy, Aya Moustafa Aboutaleb, Mohamed Ayman Ebrahim, Imene Maallem, Khaled Tarek Dardeer, Hamed Amr Heikal, Hazem Maher Gawish, Jana Zschüntzsch

**Affiliations:** 1Department of Neurology, RWTH Aachen University, Pauwels Street 30, 52074 Aachen, Germany; rabdelnaby@ukaachen.de; 2Faculty of Medicine, Cairo University, 1 Gamaa Street, Cairo 12613, Egypt; Khaled74ashraf@gmail.com (K.A.M.); anas.elgenidy@gmail.com (A.E.); youseftarek15@gmail.com (Y.T.S.); mahmoudbedewy11@gmail.com (M.M.B.); mohdaymanebrahim@gmail.com (M.A.E.); khaledegy99@gmail.com (K.T.D.); Hamedhikel18@gmail.com (H.A.H.); Hazem.maher.gawish@gmail.com (H.M.G.); 3Faculty of Medicine, Zagazig University, Zagazig 44519, Egypt; ayam78801@gmail.com; 4Faculty of Medicine, Pharmacy Department, University Badji Mokhtar Annaba, Zaafrania Street, Annaba 23000, Algeria; imene.maallem@yahoo.com; 5Clinic for Neurology, University Medical Center Göttingen, Robert-Koch-Straße 40, 37075 Göttingen, Germany

**Keywords:** muscle ultrasound, inclusion body myositis, meta-analysis, idiopathic inflammatory myopathy, neuromuscular disorder

## Abstract

Inclusion body myositis (IBM) is a slowly progressive muscle weakness of distal and proximal muscles, which is diagnosed by clinical and histopathological criteria. Imaging biomarkers are inconsistently used and do not follow international standardized criteria. We conducted a systematic review and meta-analysis to investigate the diagnostic value of muscle ultrasound (US) in IBM compared to healthy controls. A systematic search of PubMed/MEDLINE, Scopus and Web of Science was performed. Articles reporting the use of muscle ultrasound in IBM, and published in peer-reviewed journals until 11 September 2021, were included in our study. Seven studies were included, with a total of 108 IBM and 171 healthy controls. Echogenicity between IBM and healthy controls, which was assessed by three studies, demonstrated a significant mean difference in the flexor digitorum profundus (FDP) muscle, which had a grey scale value (GSV) of 36.55 (95% CI, 28.65–44.45, *p* < 0.001), and in the gastrocnemius (GC), which had a GSV of 27.90 (95% CI 16.32–39.48, *p* < 0.001). Muscle thickness in the FDP showed no significant difference between the groups. The pooled sensitivity and specificity of US in the differentiation between IBM and the controls were 82% and 98%, respectively, and the area under the curve was 0.612. IBM is a rare disease, which is reflected in the low numbers of patients included in each of the studies and thus there was high heterogeneity in the results. Nevertheless, the selected studies conclusively demonstrated significant differences in echogenicity of the FDP and GC in IBM, compared to controls. Further high-quality studies, using standardized operating procedures, are needed to implement muscle ultrasound in the diagnostic criteria.

## 1. Introduction

Inclusion body myositis (IBM) is one of the most common subtypes of idiopathic inflammatory myopathies (IIM) [1], primarily affecting those above 45 years of age. It has a progressive course and affects skeletal muscles with a distinct pattern, [2] causing asymmetric muscle weakness [3,4,5] mainly in finger flexors and/or quadricep muscles [6]. This combination of weakness often results in loss of ambulation and independence, as well as the need for assertive devices and increased supportive care over the duration of the disease. Dysphagia is also frequent in patients, which leads to swallowing disorders and an increased risk of aspiration pneumonia [7,8]. The prevalence of IBM was estimated to be between 24.8–46 patients per 1,000,000, with a male to female ratio of 2:1 [9,10]. Overall life expectancy is still shorter in IBM patients compared to the general population [11]. Although the pathophysiology of IBM has not yet been clearly elucidated [12], several factors, e.g., genetic, aging, immunologic and mitochondrial dysfunction, have been suggested to play a role [13,14,15,16,17,18,19,20].

Due to the insidious character and slow progression of the disease, there is often an estimated delay between 3–5 years in reaching a diagnosis of IBM, [13,14,15,16,17,18,19,20,21], which makes the diagnosis difficult in the early stages [22]. Currently, the diagnosis is based on clinical presentation, CK-values and histopathological findings, such as simultaneous presence of inflammatory and degenerative changes [23]. Antibodies against cytosolic 5’-nucleotidase (cN) -1A (anti-5NT1A/5NTC1A; alternatively: muscle protein 44; Mup44) can be present in 30–80% of all IBM patients and is often associated with more severe dysphagia [24].

Nevertheless, the diagnosis of IBM may be challenging as it might interfere with other subtypes of myopathies [21]. Hence, developing alternative and efficient diagnostic techniques became a necessity for assuring early diagnosis of the disease. Muscle sonography has been described as an emerging tool for diagnosis of many muscle affections and their characteristic patterns [25,26], as well as for evaluating several inflammatory neuropathies [27,28].

Therefore, the aim of this study was to evaluate the current evidence and effectiveness of the muscle ultrasound (US) as a reliable method to investigate and diagnose IBM, as well as validating its sensitivity and specificity regarding its performance in improving patient management.

## 2. Materials and Methods

We performed a systematic search on the following databases: Medline (through PubMed), Scopus and Web of Science, up until 11 of September 2021. We used the following search strategy and we searched on the previously mentioned databases using the “title and abstract“ domain in order to reach all the studies discussing the use of ultrasound to measure the muscle parameters in inclusion body myositis (IBM). The synonyms of our search strategy were retrieved from the MeSH database and were revised by a senior author (R.A). The terms were combined by “OR” and “AND” Boolean operators according to the method described in the Cochrane Handbook for Systematic Reviews of Interventions (Chapter 4.4.4) [29] as follows: (Ultrasonography OR ultrasound OR Ultrasonic OR Echotomography OR Sonography OR Sonographic OR Ultrasonographic OR Echography OR Ultrasonic) AND (Inclusion Body Myositis OR Inclusion Body Myositis OR Inclusion Body Myopathy). Our study was performed in accordance with the Preferred Reporting Items for Systematic Reviews and Meta-Analyses (PRISMA) guidelines [30]. The details of the search process, and the included studies, are shown in the following PRISMA diagram (Figure 1).

### 2.1. Selection Criteria

We included the full-text articles published in peer-reviewed journals that measured muscle parameters, such as muscle thickness and muscle echo intensity by ultrasound, in IBM patients diagnosed by muscle biopsy, and compared them to healthy controls. We also included studies that calculated the sensitivity and specificity of ultrasound in the diagnosis of IBM for the diagnostic test accuracy analysis. We excluded case reports, letters to the editor and studies that did not provide numerical data for muscle echo intensity or muscle thickness.

### 2.2. Selection and Screening

Four authors (A.E., A.M., M.B., Y.T.S.) screened the articles by title and abstract, two authors (A.M., M.B.) independently screened the articles by reading their full text and a third author (Y.T.S.) was referred to in case of a disagreement. Two authors (A.M. and M.B.) extracted data regarding patient characteristics, US devices used and outcome measures (muscle echo intensity and muscle thickness).

### 2.3. Quality Assessment

Two authors (M.B. and Y.T.S.) evaluated the quality of the included studies using the National Institute of Health (NIH) quality assessment tool for observational cohort and cross-sectional studies [31]. This tool consists of 14 questions regarding the sample size and selection and exposure and outcome assessment. Studies scoring ≥9 points were considered of good quality, 5–8 points of fair quality and 1–4 points of poor quality. Diagnostic accuracy studies were assessed using Cochrane’s QUADAS-2 Tool.

### 2.4. Statistical Analysis

We conducted two types of analyses: double arm meta-analysis and diagnostic test accuracy analysis. We used review manager software version 5.4 (The Cochrane Collaboration, UK) [32] and OpenMetaAnalyst software [33] for the double arm analysis. This analysis pooled the mean difference between IBM patients and healthy controls in terms of muscle echo intensity and muscle thickness. The random effects model of the DerSimonian Laird method was applied to account for heterogeneity between the studies [34]; *p* < 0.05 was considered statistically significant. Heterogeneity was assessed using the Chi-squared test and the I^2^ statistic; *p* < 0.05 proved significant heterogeneity and I^2^ > 50% indicated substantial heterogeneity. Leave-one-out sensitivity analysis was carried out to examine the effect of omitting each study on the overall result. Publication bias could not be assessed using funnel plots due to the small number of included studies [29].

We used Meta-DiSc software [35] for the diagnostic test accuracy analysis to pool the sensitivity and specificity, as well as the likelihood ratios of ultrasonography, in the diagnosis of IBM compared to the other reference tests (muscle biopsy and magnetic resonance imaging (MRI)). A receiver–operator curve (ROC) was created and the area under the curve (AUC) was calculated to evaluate the performance of US as a diagnostic test for IBM.

## 3. Results

### 3.1. Study Characteristics

A total of 7 studies [25,36,37,38,39,40,41], with 108 IBM patients and 171 healthy controls, were included in the analysis. Four studies [25,36,38,41] reported the echogenicity of the flexor digitorum profundus (FDP) and gastrocnemius (GC) muscle groups with 168 participants (76 IBM patients and 92 controls). Two studies [38,39] reported the muscle thickness of the FDP with 128 participants (46 IBM patients and 82 controls). Three studies [36,37,40] were involved in diagnostic test accuracy meta-analysis. The characteristics of the included studies are presented in Table 1.

### 3.2. Quality Assessment

The quality of included studies was assessed using the NIH scale and QUADAS-2 tool. Using the NIH scale, one study scored 9 and was considered of high quality, while four studies were considered to be of fair quality (score 5–8) (Table 2). For diagnostic test accuracy studies, some concerns regarding the patient selection methodology were found for two studies [36,37], while no concern regarding applicability was found for other domains (Figure 2).

### 3.3. Echogenicity of FDP

Three studies [25,36,38] assessed the FDP muscle echogenicity using ultrasound between 70 IBM (129 US measurements) cases and 92 controls (173 US measurements). Gray scale value (GSV) was graded using an arbitrary unit that ranged from 0 to 255, from black to white [36,38]. Echogenicity index mean difference was 36.55 GSV (95% CI, 28.65–44.45, *p* < 0.001), thus the echogenicity index was significant and higher in IBM patients than in controls. Between-study heterogeneity was high and significant (I^2^ = 70%, *p* = 0.02) **(**Figure 3).

A study by Noto et al. in 2013 (six patients), assessed muscle echo intensity using the Heckmatt rating scale (1–4) for two muscles: the FCU and FDP. The Heckmatt rating scale mean (range) was 1.3 (1-3) for the FCU and 2.7 (2-3) for the FDP. These data were not included in the GSV analysis.

### 3.4. Echogenicity of GC

Ultrasound analysis between 70 IBM patients (129 US measurements) and 98 controls (185 US measurements) in three studies found that the mean difference in echogenicity index was 27.90 GSV (95% CI 16.32–39.48, *p* < 0.001), denoting significance and a higher value in IBM patients than in healthy controls in the GC. Significant heterogeneity was found between study data (I^2^ = 84%, *p* < 0.001) (Figure 4).

Nodera et al. established higher echoic signal in the GC and FDP of IBM patients [36]. Subsequently, Albayda et al. found a significant difference in echogenicity between IBM patients and the control group, with FDP being the most discerning [25]. Furthermore, Leeuwenberg et al. assessed the echogenicity and muscle thickness in a group of 41 IBM patients with variable disease duration; Radboudumc 67.2 (12–228) months and Johns Hopkins 116.9 (30–360) months, as reported per median and range [38]. It showed a drastic surge in echogenicity in IBM patients.

### 3.5. Muscle Thickness of FDP

Analysis of two studies between 46 IBM cases (87 US measurements) and 82 controls (135 US measurements) revealed no significant difference in muscle thickness between IBM patients and healthy controls (MD = −1.75, 95% CI −5.01–1.51, p = 0.29). Heterogeneity was significant and high (Appendix A).

Regarding the muscle thickness of FDP, the study from John Hopkins presented a significant increase in the thickness compared to the control group. However, Radboudumc’s data included the point of no difference, denoting no statistical difference. Paramalingam et al. compared a group of five IBM patients to a group of age- and sex-matched healthy controls. The results showed no significant association between the muscle thickness of the FDP in IBM patients and in the control [39].

Noto et al., in 2013, also assessed muscle atrophy by measuring the muscle CSA. The mean CSA (range) was 80.5 mm^2^ (63.0–117.4) for the FDP muscle and 131.8 mm^2^ (109.0–149.5) for the FCU muscle.

### 3.6. Ultrasound Diagnostic Accuracy

Three studies assessed the diagnostic performance of ultrasound, differentiating between IBM (65 US measurements) and healthy controls (41 US measurements). The pooled sensitivity and specificity of US were 0.82, 95% CI 0.75–0.88 and 0.98, 95% CI 0.89–1.00, respectively (Figure 5 and Figure 6). Between-study heterogeneity was not significant for either measurement. The pooled positive likelihood ratio, negative likelihood ratio and diagnostic odds ratio were 16.89, 95% CI 4.32–66.01, 0.21, 95% CI 0.13–0.33 and 73.88, 95% CI 15.58–350.48, respectively. The Cochran’s Q and I^2^ tests showed no significant heterogeneity between studies (Figure 7, Figure 8 and Figure 9). As for the summary ROC curve (SROC)**,** the area under the curve (AUC) was 0.612 and the Q* index was 0.5848 (Appendix A).

### 3.7. Sensitivity and Subgroup Analysis

According to the leave-one-out sensitivity analysis, none of the outcomes were affected following the removal of a single study (Figure 10, Figure 11 and Figure 12). A relatively large, but insignificant, change was found in muscle thickness difference outcome following removal of Leeuwenberg (the John Hopkins population) [38]; the overall mean difference following its removal was still insignificant between patients and controls (Figure 12). We also performed sensitivity analyses to examine the effect of the study design (retrospective or prospective) on the heterogeneity in the meta-analyses. The study by Leeuwenberg et al. [38] was found to have a high impact on heterogeneity, as heterogeneity dropped substantially after eliminating it from the analysis, especially the Leeuwenberg (Radboudumc) population in the FDP echogenicity analysis (Appendix A) or the Leeuwenberg (John Hopkins) population in muscle thickness meta-analysis (Appendix A), and there was no major impact on the pooled effect after their elimination. However, there was only a minimal change in the pooled effect and heterogeneity in the GC echogenicity meta-analysis after leaving it out of that study [38].

To assess the impact of disease duration on the overall effect and heterogeneity, we conducted subgroup analysis of the FDP and GC echogenicity meta-analysis across studies by mean duration of disease, either < 70 months or ≥70 months. The result of the subgroup analysis showed that there was no statistically significant difference in overall FDP and GC echogenicity between the subgroups (*p* = 0.77, I^2^ = 0%; *p* = 0.59, I^2^ = 0% respectively) and the pooled effects of FDP and GC echogenicity analysis for each subgroup were consistent with the primary analysis (Appendix A).

### 3.8. Heterogeneity

Substantial heterogeneity was observed among the studies (I^2^ = 70% and I^2^ = 84%) with regards to the echogenicity of the FDP and GC muscle group set, respectively. Moreover, muscle thickness demonstrated higher heterogeneity (I^2^ = 91%). Regarding the diagnostic test accuracy meta-analysis, the heterogeneity in the sensitivity and the specificity analysis recorded X^2^ = 1.7; *p* = 0.42 and X^2^ = 2.42; *p* = 0.49, respectively. The threshold analysis for different cut-off values was made using Spearmann’s Test in an attempt to identify the reason for heterogeneity, and yielded no effect. Only three [36,37,39] out of the seven studies [25,36,37,38,39,40,41] evaluated the inter-rater or intra-rater reliability of the measurements. Accordingly, publication bias assessment could not be performed due to the small number of included studies.

Following the strategies described by Part Two of the Cochrane Handbook 9.5.3 for addressing the heterogeneity, we performed a meta-regression analysis for covariates that may have potentially caused heterogeneity. The meta-regression showed that the US device could be a source of heterogeneity in the meta-analysis, comparing the FDP muscle echogenicity between IBM patients and controls (*p*-value 0.01 and 0.03), as shown in Appendix A. However, the meta-regression analyses for other analyses showed that the device was not a source of heterogeneity. Meta-regression may be not conclusive when there are few studies included in the meta-analysis.

In terms of the laterality of muscle evaluation, four studies [25,37,38,40] used the average of bilateral examination. It is worth noting that Leeuwenberg (John Hopkins) [38] used the average of the bilateral examination while Leeuwenberg (Radboudumc) [38] mostly used the average of the bilateral examination, and in cases of unilateral examination, these results were used as representative for both sides. Five studies [25,36,38,39,41] used the quantitative method for the scoring system. Three of them [36,39,41] also reported using the Heckmatt rating scale (which is a semi-quantitative scale), which included the use of a modified Heckmatt rating scale by one study [36], whereas, two studies [37,40] used only the Heckmatt rating scale.

## 4. Discussion

The results of our meta-analyses demonstrated an increased echogenicity in certain muscle groups compared to controls and could therefore be used as a supportive criterion in the challenge of an early diagnosis for IBM. In detail, the echogenicity in IBM patients was increased with 36.55 GSV, 95% CI 28.65–44.45 and 27.9 GSV 95% CI, 16.32 −39.48 for the FDP and GC muscle groups, respectively, and the test of overall effect was significant for both measurements (*p* < 0.001). In contrast, results of muscle thickness in IBM patients showed no difference with −1.75, 95% CI, −5.01 to 1.51. No potential outliers were identified by conducting leave-one-out sensitivity analysis. The heterogeneity testing was significant in the measures of echogenicity in the FDP and GC muscle groups, muscle thickness in the FDP and diagnostic test accuracy. Results of diagnostic test accuracy meta-analysis revealed a collective specificity of 0.98, 95% CI, 0.89–1 and sensitivity of 0.82 CI 95%, 0.75–0.88. This agreed with the data of a previous study that showed significant accuracy of US compared to MRI in detecting muscle abnormalities and fat infiltration of IBM patients [40]. On the basis of a literature comparison, US provided a higher sensitivity than anti-cN1A-antibody testing, a less invasive technique than electromyography (EMG) and muscle biopsy, as well as a less expensive method than MRI.

The anti-cN1A-antibody sensitivity greatly varied in several published studies; ranging from 33% to 76% [24,42], while specificity was indicated from 87% to 100%. The anti-cN1A-antibody was also detected in other IIM and autoimmune diseases [43,44].

Needle EMG is a well-established and historically used neurophysiological test, for which a sensitivity of 89% was determined for IIM in the work of Bohan and Peter [45] and abnormalities of IIM were described in more detail by [46]. For IBM, the data were even less frequent and showed, e.g., spontaneous activity in 62.5% [47] or classical myopathic changes [48] in EMG examinations. Recently, diagnostic accuracy has been evaluated for IIM, demonstrating a sensitivity of 85.2% in IBM [49].

Muscle biopsy is the diagnostic gold standard for identifying the typical pathological features with a limited sensitivity in early disease phases, which may necessitate multiple muscle biopsies to establish a diagnosis [50]. However, the full pathological picture (rimmed vacuoles, p62 aggregates, increased major histocompatibility complex class I expression and endmysial T cells) reached a sensitivity of 93% and specificity of 100% [50].

To increase diagnostic certainty as early as possible, a non-invasive imaging technique could be used as a supportive diagnostic measure with the potential of repeated application and without potential harm. Both MRI and US fulfil these criteria. Until now, MRI remains the most widely used technique for muscle imaging as such modality can delicately visualize the distribution of affected muscles and surrounding tissues (fascia and skin), disease activity and permanent muscle damage, such as muscle atrophy and/or replacement by fatty tissue [51]. For IBM, the sensitivity of thigh muscle MRI was evaluated in 2002 and showed 72% [52]. Whole-body MRI also contributed to specific recognition patterns in IBM [10], e.g., with a sensitivity of 80% and specificity of 100% for morphological/degenerative features of the quadriceps femoris muscle [53]. Thus, these two statistical parameters are comparable to US, whereby the MRI application has the advantage of less operator dependency but also some potential drawbacks, such as its high cost, time-consuming nature (whole-body MRI requires approximately 1 h), lack of widespread availability and exclusion of subjects with metal implants, pacemakers or claustrophobia. The more cost-effective, faster, geographically widespread (e.g., in rural areas and for patients with restricted mobility), bedside and alternative method is US, with its excellent image resolution and the ability to detect morphological changes in the muscle, such as edema, atrophy and muscle replacement by fibroadipose tissue. The visualization of these myocharacteristics by the US parameters of muscle echogenicity and muscle thickness [36] have led to this method also being used for other myopathies and as follow-up assessment for disease severity and residual muscle damage [40].

In detail, echogenicity is a hallmark feature of muscle replacement by fatty tissue and fibrosis [54] and is widely accepted as an ultrasound parameter in chronic muscle pathologies, such as myositis [55,56,57]. Mainly three methods are available to assess muscle echogenicity: (1) visual qualitative method to determine echogenicity in relation to other tissues; (2) semiquantitative evaluation by a scale [58]; and (3) grayscale-based quantitative measurement. Independent of the 40-year long availability, different evaluation methods for echogenicity are used in clinical trials, indicating a further need of harmonizing and combining them.

Standardization of muscle thickness is another issue to be solved, as various factors, such as body size of the patient, disease duration, sex and exact anatomical localization, might affect measurements [38,39].

Beside the patient-dependent factors leading to the heterogeneity in our meta-analysis, another major source is the dependency on the ultrasound operator primed with the anisotropic nature of the muscle, in which any trace change in the viewing angle alters the echo intensity. However, to overcome this, Scott et al. recommended taking extra care to reduce the probe tilt, especially while evaluating the echo-intensity values [57].

Other probable causes of heterogeneity include the variation in sampling, clinical characteristics of IBM patients and the differences in the control groups.

To our knowledge, this is the first systematic review and meta-analysis to convey the echogenicity and muscle thickness in IBM and to evaluate the sensitivity and specificity of ultrasonography to support the diagnosis of IBM by conducting diagnostic test accuracy meta-analysis. Although our literature searches were thorough and data extraction were cautious, limitations still exist in this review, such as the small number of published papers on IBM. However, as suggested by Callan et al., periodical evaluation of the current evidence is mandatory to increase awareness and improve research methodology [9]. Further meta-analysis is considered once additional homogenous studies become available. In addition, upcoming studies on the diagnostic value of muscle ultrasonography should include larger samples of patients with IBM, as current evidence of sonography is not on a sufficient standardized level for the diagnosis of IBM. Nevertheless, we propose that the deployment of US, in combination with the standard clinical, histological and serological assessment, and with the long-term goal of its implementation, adds value to education and clinical practice.

Finally, the development of international standard operating procedures, e.g., as developed for measures of disease activity in myositis by iMACS [58], would equalize the imaging evaluation and provide the basis for longitudinal studies. Advances in artificial intelligence will further facilitate US as a useful diagnostic and follow-up/longitudinal technique [59], which will foster its use in clinical trials [60].

## Figures and Tables

**Figure 1 cells-11-00600-f001:**
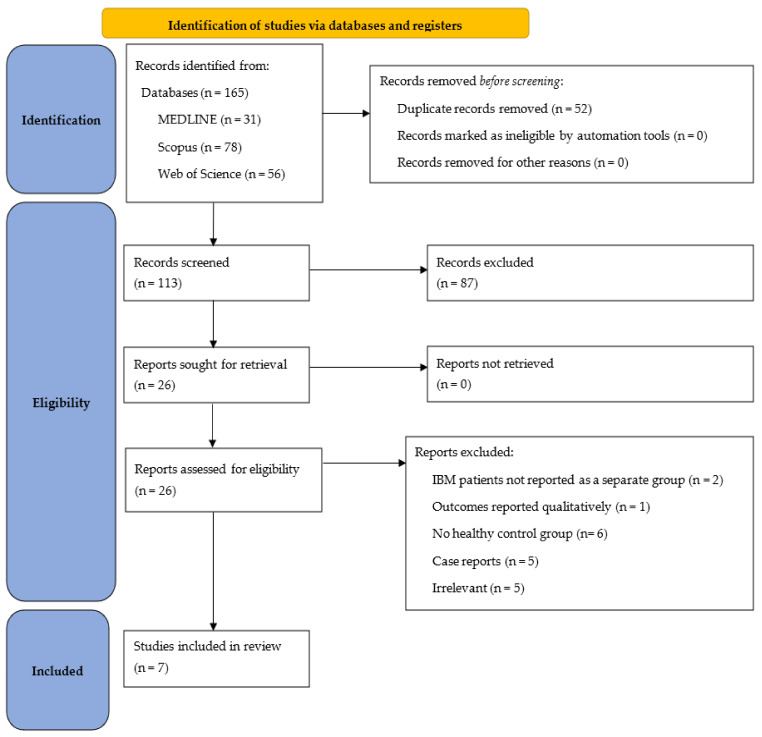
PRISMA flow chart detailing the search process and studies included.

**Figure 2 cells-11-00600-f002:**
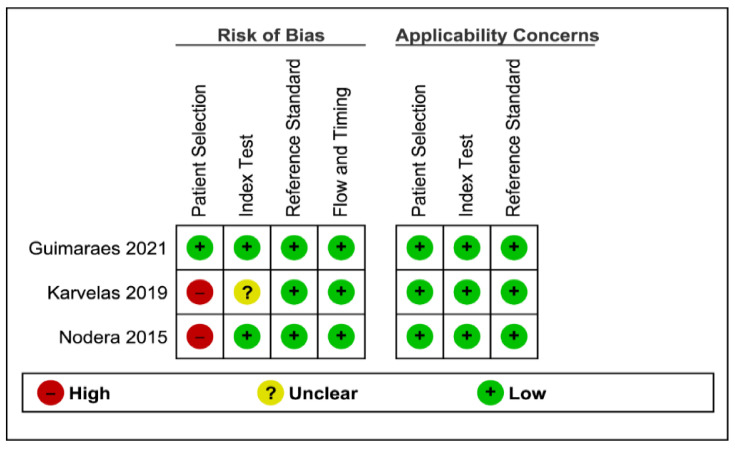
Final methodological quality summary.

**Figure 3 cells-11-00600-f003:**
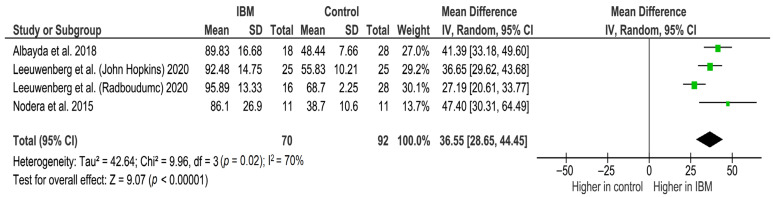
Forest plot of the echogenicity of FDP in grey scale value (GSV) of the included studies.

**Figure 4 cells-11-00600-f004:**
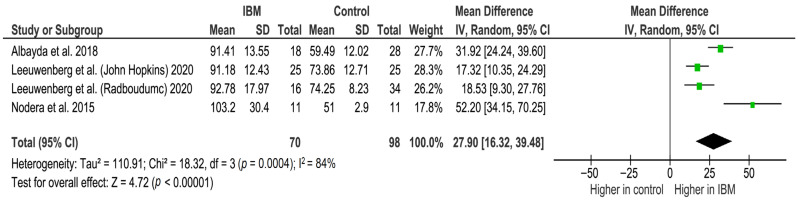
Forest plot of the echogenicity of the GC muscle in GSV in the included studies.

**Figure 5 cells-11-00600-f005:**
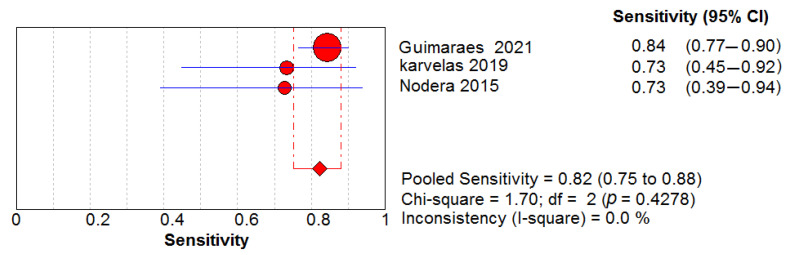
Sensitivity of the US in the diagnosis of IBM.

**Figure 6 cells-11-00600-f006:**
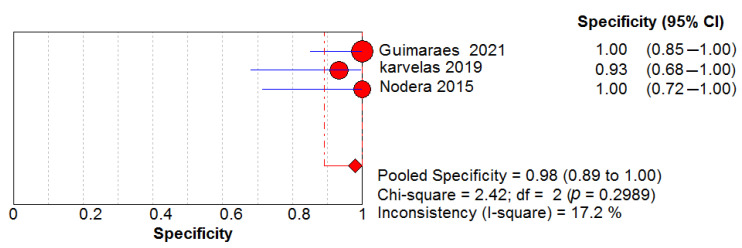
Specificity of the US in the diagnosis of IBM.

**Figure 7 cells-11-00600-f007:**
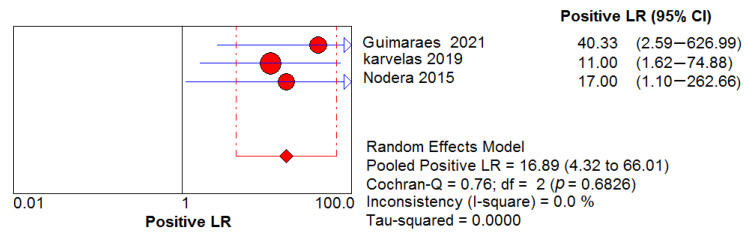
Positive likelihood ratio (LR) shows no significant heterogeneity between studies.

**Figure 8 cells-11-00600-f008:**
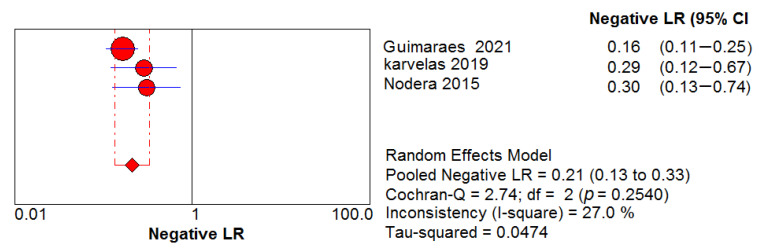
Negative LR shows no significant heterogeneity between studies.

**Figure 9 cells-11-00600-f009:**
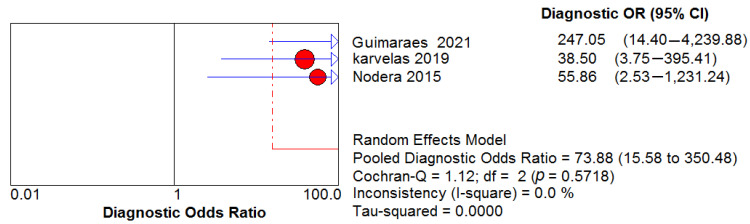
Diagnostic OR shows no significant heterogeneity between studies.

**Figure 10 cells-11-00600-f010:**
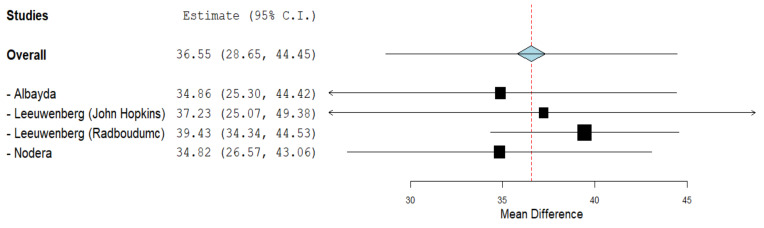
Leave-one-out analysis of the studies commented on the echogenicity of the FDP muscle.

**Figure 11 cells-11-00600-f011:**
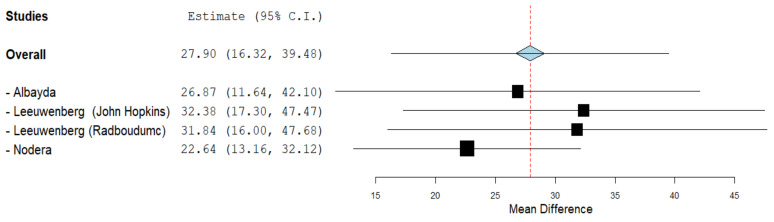
Leave-one-out analysis of the studies commented on the muscle thickness of the GC muscle.

**Figure 12 cells-11-00600-f012:**
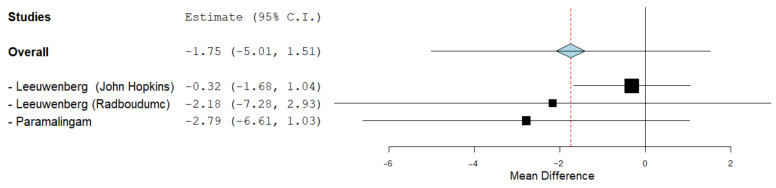
Leave-one-out analysis of the studies commented on the muscle thickness of the FDP muscle.

**Table 1 cells-11-00600-t001:** Characteristics of the included studies.

							IBM				Control		
Author	Year	Country	Study Design	US Device Type	Evaluated Muscles (Echogenicity +/− Thickness)	Bilateral Muscle Evaluation	No. of Patients	Mean Age in Years (SD) {Range}	Sex M:F	Duration of Disease, Mean (SD)	No. of Patients	Mean Age in Years (SD) {Range}	Sex M:F
Albayda	2018	USA	Prospective	GE Logiq e with 12L linearphased array transducer	Muscle Echointensity: FDP, flexor carpi ulnaris (FCU), GC, deltoids, biceps, rectus femoris (RF) and tibialis anterior (TA)	Yes	18	64.8 (9.8) {52–84}	9:9	138.0 (93.2)	28	48.5 (15.5) {23–74}	13:15
Nodera	2015	Japan	Prospective	LOGIQ7 with a fixed 11-MHz linear-array transducer	Muscle Echointensity: medial head of the GC, soleus, FDP and FCU	No, only right side chosen in order to avoid potential selection bias of highly correlated, bilateral data from the same individual	11	74.5 (6.8) {62–82}	7:4	48.0 (40.2)	11	73.5 (9.9) {57–88}	6:5
Paramalingam	2021	Australia	Prospective	Canon Aplio 500 machine with a 14-5 linear probe set at 14 MHz	Muscle Echointensity and thickness: FDP, FCU, vastus lateralis (VL), TA and deltoid	No (on the participant’s right-hand side)	5	70.40 (6.50)	5:0		29	46.60 (16.10)	16:13
Leeuwenberg (John Hopkins)	2020	USA and Netherlands	Retrospective	GE Logiq e with 12L linearphased array transducer	Muscle Echointensity and thickness: FDP, medial head of the GC, RF and/or VL	Yes	25	65.1 {52–80}	12:13	116.9 {30–360}	25	65.9 {51–80}	9:16
Leeuwenberg (Radboudumc)	2020	USA and Netherlands	Retrospective	Esaote Mylab Twice machine (Esaote SpA, Genoa, Italy) with an LA533 linear 3- to 13-MHz transducer	Muscle Echointensity and thickness: FDP, medial head of the GC, RF and/or VL	Mostly but unilateral results were used as representative of both sides	16	70.5 {54–84}	9:7	67.2 {12–228}	63	63.4 {50–82}	28:35
Karvelas	2019	USA	Prospective, blinded	GE Logiq S8 (GE Healthcare, Chicago, Illinois) with a 15-MHz linear array transducer; an Esaote MyLabGamma (Esaote S.p.A, Genoa, Italy) with an 18-MHz linear array transducer; and a Biosound MyLab25 (Esaote) with an 18-MHz linear array transducer.	FDP and FCU	Yes	15	73.2 (5.78) {61–83}	13:2		15	55.93 (13.82) {34–79}	7:8
Guimaraes	2021	Brazil	Prospective	HD II XE (Philips Medical Systems, Nederland B.V., Best, Netherlands) US system with a 12-MHz linear array	Quadricep muscle group (RF, vastus medialis, vastus intermedius, VL), GC and FDP	Yes	12	63.3 (9.1) {49–84}	11:1	90.6 (35.1)			
Noto	2013	Japan	Prospective	GE Logiq P5 system with a 10-MHZ linear-array probe (GE Healthcare Japan)	Echogenicity in FDP–FCU	no	6	71.5 {68–79}	5:1	56.7 {14–120}			

**Table 2 cells-11-00600-t002:** Quality assessment of the included studies according to the National Institute of Health (NIH) quality assessment tool for observational cohort and cross-sectional studies.

Study	C 1	C 2	C 3	C 4	C 5	C 6	C 7	C 8	C 9	C 10	C 11	C 12	C 13	C 14
Paramalingam 2021	Yes	Yes	CD	Yes	No	NA	Yes	NA	Yes	NA	Yes	Yes	Yes	Yes
Leeuwenberg 2020	Yes	Yes	Yes	Yes	No	NA	Yes	No	Yes	No	Yes	No	Yes	No
Albayda 2018	Yes	Yes	Yes	Yes	No	NA	Yes	No	Yes	No	Yes	No	Yes	No
Nodera 2015	Yes	No	CD	Yes	No	NA	Yes	No	Yes	No	Yes	Yes	Yes	No
Noto 2013	Yes	Yes	CD	Yes	No	NA	Yes	NA	Yes	NA	Yes	No	Yes	No

C—criterion; CD—cannot be determined; NA—not applicable; Criterion 1—clear statement of research question and objective; Criterion 2—clear specification and definition of the study population; Criterion 3—participation rate of 50% of eligible persons; Criterion 4—selection of subjects from the same population and time period and application of selection criteria on all subjects uniformly; Criterion 5—justification of sample size; Criterion 6—measurement of exposure before measurement of outcome; Criterion 7—sufficient timeframe to predict an association between exposure and outcome; Criterion 8—examination of different levels of the exposure in relation to the outcome; Criterion 9—clear and valid definition of exposure measures and its consistent implementation on study subjects; Criterion 10—assessment of exposure more than once over time; Criterion 11—clear and valid definition of outcome measures and its consistent implementation on study subjects; Criterion 12—blinding of outcome assessors to exposure status of participants; Criterion 13—loss to follow up being 20% or less; Criterion 14—measurement or statistical adjustment of confounding variables.

## Data Availability

The data presented in this manuscript are available from the corresponding databases: Medline (through PubMed) (https://pubmed.ncbi.nlm.nih.gov/ (accessed on 11 September 2021)), Scopus (http://scopus.com/home.uri (accessed on 11 September 2021)) and Web of Science (https://www.webofscience.com (accessed on 11 September 2021)).

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
