# Peer review of "Muscle Sonography in Inclusion Body Myositis: A Systematic Review and Meta-Analysis of 944 Measurements"

_cells, 2022, doi:10.3390/cells11040600_

Round 1

Reviewer 1 Report

This is a meta-review of the muscle ultrasound findings in IBM compared to controls. It is an interesting analysis that has value for the neuromuscular community dealing with this type of patients.

The paper is well-written overall, and only in the later paragraphs (such as 3.8) the English needs a little editing here and there.

While the use of a group mean Heckmatt scale is understandable (in both the original studies as in this current meta analysis), it does not make sense from a methodologic standpoint, as Heckmatt grading is an ordinale scale with no linear or numeric relation between the different classes. Please consider and discuss.

In the Results paragraph 3.6, please remove the brackets around the primary data results of 0.82, 0.98, respectively, and similarly do so for the likelihood ratios.

Author Response

Dear reviewer,

We have read the comments and questions carefully and thank you for your the detailed suggestions, which we have amended or implemented both in the manuscript as well as in the attached point-to-point response. 
We hope that we have answered all points adequately.

Kind regards

Reviewer 2 Report

This paper summarizes a statistical evaluation of suitable papers for a meta-analysis of the accuracy of ultrasound in the diagnosis of inclusion body myositis.  The authors document the proper steps for conducting this type of analysis in the methods, the results are consistent with the discussion and the introduction is appropriate.

  The authors may wish to consider expanding the discussion, for the sake of perspective, to briefly compare the accuracy of ultrasound with other available diagnostic assessments.  For example reference 40 provides evidence that ultrasound compares favorably with MRI, a more expensive and difficult technique.  Other studies have looked at the accuracy of antibody testing which is specific but not as sensitive and electrodiagnostic testing which is sensitive but not as specific.  Readers are likely interested in the utility of the technique compared to other options and this might enhance the likelihood of future citations of this manuscript. 

Author Response

Dear reviewer,

We have read the comments and questions carefully and thank you for your suggestions, which we have amended or implemented both in the manuscript and in the attached pint-to-point response. 
We hope that we have answered all points adequately.

Kind regards
